# Digital and Navigational Health Literacy in Surgical Patients: Vulnerabilities in the Transition to Post-Discharge Care

**DOI:** 10.3390/healthcare13243227

**Published:** 2025-12-10

**Authors:** Patrícia Valentim, Miguel Arriaga, Paulo Nogueira, Andreia Costa

**Affiliations:** 1Nursing Research Innovation and Development Centre of Lisbon (CIDNUR), School of Nursing, University of Lisbon, 1600-190 Lisbon, Portugal; 2Directorate-General of Health (DGS), Alameda D. Afonso Henriques, 45, 1049-005 Lisboa, Portugal; 3Institute of Environmental Health, ISAMB, Faculty of Medicine, Av. Prof, Egas Moniz, 1649-028 Lisbon, Portugal; 4Laboratory for Land Use and Ecosystem Services Sustainability—TERRA, 1349-017 Lisbon, Portugal

**Keywords:** digital health literacy, navigational health literacy, surgical patient, care transition, hospital discharge

## Abstract

**Highlights:**

**What are the main findings?**
Among 94 surgical inpatients, 25–27% showed inadequate digital or navigational health literacy.Digital health literacy declined significantly across age groups (*p* = 0.038), while navigational literacy varied widely without age-related differences (*p* = 0.239).Emergency surgery was independently associated with lower navigational literacy (β = −31.07; *p* = 0.018).Digital and navigational literacy were strongly correlated (ρ = 0.86), indicating overlapping competencies.

**What are the implications of the main findings?**
As healthcare increasingly shifts toward digital models, deficits in digital and navigational literacy may exacerbate inequities and hinder safe post-discharge transitions.Developing age-adapted and context-sensitive strategies—particularly for patients undergoing emergency surgery—is essential to promote equity and prevent digital exclusion.

**Abstract:**

Background/Objectives: The digitalization of healthcare in general, and particularly of surgical care, increasingly requires patients to navigate online information and digital follow-up systems. Limited digital or navigational health literacy may hinder safe recovery and exacerbate health inequities. This study aimed to assess levels of digital and navigational health literacy and their associations in a sample of surgical patients. Methods: A cross-sectional study was conducted with ninety-four adults undergoing elective or emergency surgery, using the validated HLS19-DIGI (digital health literacy) and HLS19-NAV (navigational health literacy) instruments. Data collection took place between March 2025, and 28 August 2025, at a hospital in the Lisbon district. Descriptive analyses, bivariate analysis, and multiple regression were used to identify sociodemographic and clinical predictors. Results: Digital literacy varied significantly across age groups (*p* = 0.038), with median scores declining from 87.5 (31–45 years) to 31.2 (>65 years). Navigational literacy was lower in emergency versus elective surgery patients (41.7 vs. 83.3; *p* = 0.036). In adjusted models, self-employment predicted lower digital literacy (β = −36.06, *p* = 0.009), while emergency surgery remained the only predictor of navigational literacy (β = −31.07, *p* = 0.018). Digital and navigational literacy were strongly correlated (ρ = 0.859). Conclusions: The surgical patients in this study demonstrated marked literacy deficits, particularly older adults and those undergoing emergency procedures. Digital and navigational literacy appear to reflect overlapping competencies. As healthcare shifts toward digital models, it is essential to develop age-adapted strategies to promote equity and prevent exclusion.

## 1. Introduction

Transitions in surgical care increasingly require patients to engage with health information, navigate complex care pathways, and interact with digital systems. From pre-operative preparation to discharge and follow-up, modern peri-operative care relies on the ability to interpret instructions, locate and use appropriate services, and communicate effectively with healthcare professionals—tasks that depend heavily on health literacy [1]. The rapid digitalization of healthcare, accelerated by the COVID-19 pandemic, has further intensified this dependence, potentially creating new barriers for vulnerable populations, particularly older adults [2].

Health literacy is broadly defined as the ability to access, understand, appraise, and apply health information to make informed decisions [3,4]. Two domains are particularly relevant in the surgical context: digital health literacy, which refers to the ability to obtain and critically evaluate online health information and interact with digital services [5,6,7]; and navigational health literacy, which encompasses the ability to understand how the healthcare system operates, identify appropriate services, assess the quality of care, and carry out the organisational tasks required to access these services [8,9]. Both constructs are part of the conceptual framework of the European Health Literacy Survey 2019 (HLS19). Although related, each reflects distinct competencies that influence patients’ ability to manage transitions safely. Older adults, individuals with limited digital experience, and patients undergoing emergency surgery may be particularly vulnerable to literacy-related challenges [10,11].

In Portugal, the digitalization of health services has advanced rapidly through national initiatives such as the SNS Portal and SPMS e-Prescription, which enable online appointment booking, medication renewal, and access to electronic health records [12,13]. While these innovations promise greater efficiency and patient engagement, they may inadvertently disadvantage those with limited literacy or digital skills, reinforcing social inequalities in access to care [14,15]. This risk is especially pronounced among older adults, who show the lowest levels of digital health literacy and face barriers such as limited familiarity with technology, reduced cognitive processing speed, and usability challenges associated with digital platforms [7,16].

In surgical patients, health literacy plays a decisive role in outcomes, as evidence links low literacy to poorer understanding of surgical risks, lower adherence to instructions, and higher readmission rates. [17,18]. However, unlike patients with chronic conditions, who develop health-management skills over time, surgical patients must acquire and apply complex information abruptly—often while experiencing pain, stress, or cognitive fatigue. Emergency surgery adds further challenges, requiring patients to navigate unfamiliar systems under acute conditions [19].

Despite its importance, digital and navigational health literacy remain understudied in surgical populations. Most research has focused on general health literacy or chronic-disease contexts, with limited attention to the post-surgical phase, which is characterised by acute cognitive demands [20,21]. This study aimed to: (1) describe digital and navigational health literacy among hospitalised surgical patients; (2) analyse differences across sociodemographic and clinical characteristics using non-parametric methods; (3) identify independent predictors through multivariable regression; and (4) identify vulnerable subgroups requiring targeted support during care transitions.

## 2. Materials and Methods

### 2.1. Study Design and Setting

A cross-sectional observational study was conducted following the STROBE guidelines [22]. Data collection took place continuously between 17 March 2025 and 28 August 2025, and data analysis was carried out between September and October 2025. Participants were recruited through multimodal convenience sampling, using a non-probabilistic approach; the selection of the hospital was based on operational feasibility, prior ethical approval, and direct access to the surgical population required for the study, with this sampling strategy justified by the logistical conditions of the surgical service and the exploratory nature of the research. Data were collected through structured interviews administered face-to-face by trained researchers after patients had undergone surgery and were clinically stable, ensuring comprehension and completeness of responses; each interview lasted approximately 20 min. The study protocol was approved by the Ethics Committee of the Unidade Local de Saúde do Oeste (reference: 11 March 2025).

### 2.2. Setting

The study took place in a surgical ward of a public tertiary hospital serving a mixed urban and semi-urban population. The unit provides general surgery, urology and otorhinolaryngology services.

### 2.3. Measures

#### 2.3.1. Digital Health Literacy (HLS19-DIGI)

Digital health literacy was assessed using the HLS19-DIGI instrument of the European Health Literacy Survey 2019 [5]. The module contains 8 items, each rated on a four-point difficulty scale (1 = very difficult to 4 = very easy). After computing the mean raw score, values were transformed to the 0–100 HLS index, with higher values indicating better literacy. Internal consistency in this sample was excellent (α = 0.950; ω = 0.954).

#### 2.3.2. Navigational Health Literacy (HLS19-NAV)

Navigational health literacy was measured using HLS19-NAV, a 12-item module assessing individuals’ ability to interact with the healthcare system, including finding services, understanding system organisation and evaluating service quality. Items are rated on the same four-point difficulty scale and transformed to the 0–100 index. Internal consistency was excellent (α = 0.959; ω = 0.962). 

#### 2.3.3. Sociodemographic and Clinical Variables

Age (later categorised as 18–30, 31–45, 46–50, 51–65 and >65), sex (male/female), education (None/Primary, Basic, Secondary/Higher) and employment (Employed, Self-employed, Unemployed, Retired, Student, Other).

Clinical variables included surgical specialty and surgery context (elective/emergency).

Digital support assessed whether the participant had someone available to help with digital technologies (Yes/No).

### 2.4. Participants

Inclusion criteria were adults (≥18 years) admitted for elective or emergency surgery and able to participate in the interview. Exclusion criteria included cognitive impairment, unstable clinical condition or inability to communicate in Portuguese. Informed consent was obtained from all participants.

Of the 158 patients invited to participate, 94 completed the questionnaire (response rate = 59.49%). Twelve questionnaires were excluded due to incomplete data. Reasons for non-participation included early discharge (*n* = 8) and refusal (*n* = 44).

### 2.5. Statistical Analysis

All analyses were performed using jamovi 2.6 [23] and R 4.4.0 [24], with the psych [25] and car [26] packages. Since HLS19-DIGI and HLS19-NAV scores were non-normally distributed (Shapiro–Wilk *p* < 0.05), descriptive statistics are reported as median (IQR).

#### 2.5.1. Bivariate Analysis

Mann–Whitney U tests were used for binary variables (sex, surgery context, digital support).Kruskal–Wallis tests were applied to variables with >2 categories (age groups, education, employment). Significant Kruskal–Wallis tests were followed by Dunn’s post hoc tests with Bonferroni correction.Spearman’s rho correlations were calculated for associations between continuous variables, with 95% bootstrap CIs (1000 iterations).

#### 2.5.2. Regression Analysis

Separate multiple linear regressions were fitted for DIGI and NAV scores. All categorical predictors were dummy-coded, using the most frequent category as reference. Transformations of the outcomes (log, square-root, Box–Cox) were explored but did not improve model fit or residual diagnostics; thus, untransformed index scores were retained.

Regression assumptions were assessed through residual plots, Shapiro–Wilk tests, Breusch–Pagan tests, and variance inflation factors (VIF) (<5). Missing data were handled with listwise deletion. Results are reported as β coefficients, SE, 95% CI, *p*-values and model fit statistics.

Additional information is available in the Appendix A [27].

## 3. Results

### 3.1. Sample Characteristics

Participants had a median age of 52.5 years (IQR: 38–63), with a balanced distribution across age groups (14.9% aged 18–30; 23.4% aged 31–45; 9.6% aged 46–50; 28.7% aged 51–65; and 23.4% aged > 65). Most participants were male (60.6%). Education was predominantly in the Primary/None (39.4%) or Basic (36.2%) categories, and 60.6% of participants were employed. Emergency surgery accounted for 24.5% of admissions. Most participants reported access to digital support (91.5%). Full details are provided in Table 1.

### 3.2. Digital Health Literacy (HLS19-DIGI)

Digital literacy scores varied significantly by age group (Kruskal–Wallis *p* = 0.038). Median scores were highest among younger participants (18–30: 81.2; 31–45: 87.5) and progressively lower among older adults (51–65: 50.0; >65: 31.2). No statistically significant differences were observed for sex (*p* = 0.531), education (*p* = 0.188), employment status (*p* = 0.095), surgery context (*p* = 0.097), or digital support (*p* = 0.966). Full results are shown in Table 2.

### 3.3. Navigational Health Literacy (HLS19-NAV)

Navigational literacy did not differ significantly across age groups (*p* = 0.239). Median scores ranged widely, but with no clear monotonic trend. No statistically significant differences were found for sex (*p* = 0.397), education (*p* = 0.582), employment (*p* = 0.425), or digital support (*p* = 0.789). However, the surgery context was significantly associated with NAV, with emergency patients presenting markedly lower navigational literacy than elective-surgery patients (median 41.7 vs. 83.3; *p* = 0.036). Table 3 provides full subgroup values.

### 3.4. Age-Related Disparities

Age correlated negatively with digital literacy (ρ = −0.36; 95% CI: −0.57 to −0.14; *p* = 0.002) and with navigational literacy (ρ = −0.28; 95% CI: −0.51 to −0.02; *p* = 0.034), indicating a consistent decline across both domains. This pattern is consistent with the values shown in the tables, where median digital literacy scores progressively decrease from the 18–30 age group (81.2) and the 31–45 group (87.5) to the 51–65 (50.0) and >65 groups (31.2). Figure 1 illustrates this trend, showing a continuous reduction across age cohorts and a marked decline among the oldest participants. The Kruskal–Wallis test confirmed significant age-group differences for digital literacy (χ^2^ = 13.02; *p* = 0.005), but not for navigational literacy (χ^2^ = 7.06; *p* = 0.070).

### 3.5. Clinical and Contextual Predictors

The surgical context had a significant impact on health literacy. Patients undergoing emergency surgery showed lower navigational literacy than those in elective surgery, as confirmed in the bivariate analysis (medians: 41.7 vs. 83.3; *p* = 0.036) and remaining significant in the multivariable model (β = −31.07; *p* = 0.018). For digital literacy, although the median score was lower in emergency procedures (43.8) compared with elective ones (87.5), the difference did not reach statistical significance (*p* = 0.097) and did not remain significant in the adjusted model (β = −16.18; *p* = 0.121).

Healthcare utilization frequency correlated positively with age (ρ = 0.327; *p* = 0.001) and showed weak, non-significant negative associations with both digital literacy (ρ = −0.180; *p* = 0.134) and navigational literacy (ρ = −0.174; *p* = 0.192), suggesting that more frequent users of healthcare services may paradoxically have lower literacy levels.

The availability of digital support did not demonstrate a protective effect: although 91.5% of participants reported having someone to assist them with digital technologies, such support did not moderate the relationship between age and literacy (interaction *p* = 0.443) and did not produce significant differences in median DIGI (*p* = 0.966) or NAV scores (*p* = 0.789).

### 3.6. Relationship Between Digital and Navigational Literacy

The two dimensions showed an extremely high correlation (ρ = 0.859; *p* < 0.001), suggesting that they reflect largely overlapping competencies (Figure 2). Among participants who completed both scales (*n* = 55), no significant difference was observed between the scores (Wilcoxon *p* = 0.349), reinforcing the notion that digital and navigational literacy represent interconnected aspects of the same functional capacity.

### 3.7. Multivariable Analysis

Table 4 and Table 5 present the fully adjusted multiple linear regression models for digital and navigational literacy. After dummy coding and inclusion of all relevant covariates, only one predictor remained independently associated with digital literacy: being self-employed was associated with significantly lower HLS19-DIGI scores compared with employed participants (β = −36.06, *p* = 0.009). Age, sex, education, surgery context, healthcare utilisation, and digital support were not significant in the adjusted model (all *p* > 0.10). The model explained 17.7% of the adjusted variance (adjusted R^2^ = 0.177).

For navigational literacy, emergency surgery remained the only independent predictor (β = −31.07, *p* = 0.018). No sociodemographic variables, including age and education, were independently associated with NAV after adjustment. The model explained 9.9% of the adjusted variance (adjusted R^2^ = 0.099). All coefficients, confidence intervals and *p*-values are reported in Table 4 and Table 5.

## 4. Discussion

### 4.1. Principal Findings

This study identified substantial variability in both digital and navigational health literacy among surgical inpatients. Digital literacy showed a clear decline across successive age categories in the descriptive analyses, with younger adults demonstrating higher competencies and older age groups presenting progressively lower scores. Navigational literacy, although not significantly different across age groups, displayed wide variability. Importantly, however, these age-related differences did not persist in the multivariable models, indicating that the apparent association between age and health literacy reflects broader sociodemographic clustering rather than an independent age effect.

In contrast, surgery context exerted a consistent and meaningful influence: patients undergoing emergency surgery showed significantly lower navigational literacy, a relationship that remained robust after full adjustment. These findings reinforce the idea that vulnerabilities in health literacy arise not only from individual characteristics but also from the contextual demands of the care pathway.

Overall, the patterns observed are consistent with prior research linking limited health literacy to poorer comprehension, reduced self-management, and worse clinical outcomes in surgical care [1,17], while also highlighting a digital divide in which patients tend to cluster into high-competency or low-competency groups rather than forming a continuous gradient.

### 4.2. Apparent Versus Actual Impact of Age on Health Literacy

Although descriptive analyses revealed substantial age gradients in digital literacy and moderate age-related declines in navigational literacy, age was not an independent predictor in the adjusted models. This contrasts with population-based findings where age frequently remains a key determinant [5,8]. The disappearance of the age effect after adjustment suggests that literacy disparities across age groups may arise from cumulative inequalities—such as educational attainment, employment trajectories, or differential exposure to digital technologies—rather than chronological age alone.

This interpretation aligns with the framework of the “third-level digital divide” [7], which emphasises disparities in meaningful digital engagement even when access is widespread. While older age groups may face usability, confidence, or motivational barriers [2,16], these factors did not independently predict literacy in the multivariable analysis. Therefore, interventions should target specific vulnerability profiles rather than relying exclusively on age-based assumptions.

### 4.3. Emergency Surgery as a Determinant of Navigational Vulnerability

Emergency surgery was the only independent predictor of navigational literacy. Patients in acute pathways must navigate the health system under time pressure, heightened stress, and limited opportunity for preparation—conditions known to impair comprehension and decision-making [19]. This finding underscores the importance of adopting health literacy universal precautions [28] within emergency care, where the risk of communication mismatches is inherently higher.

Tailored educational materials, simplified communication, and structured follow-up strategies are likely essential to ensure safe transitions for patients undergoing emergency procedures.

### 4.4. Overlap Between Digital and Navigational Competencies

Digital and navigational literacy demonstrated a remarkably high correlation (ρ = 0.859), echoing findings from HLS19 population analyses where both domains load onto a shared underlying construct [10,11]. Both competencies rely on similar cognitive and procedural skills, including information appraisal, system navigation, and interaction with complex organisational structures. This overlap suggests that integrated literacy-strengthening interventions may be more effective than domain-specific approaches.

### 4.5. Implications for Surgical Care Delivery

In an increasingly digital-first healthcare landscape, disparities in digital and navigational literacy may amplify inequities in care. While most participants reported having digital support, this did not appear to mitigate literacy challenges, suggesting that external assistance cannot compensate for underlying limitations in health system navigation or digital skills.

Clinicians—particularly nurses—play a critical mediating role and should adopt literacy-sensitive communication practices, multimodal information formats, simplified digital interfaces, and involve caregivers as needed. These strategies are consistent with the principles of health-literate organisations and the WHO Global Strategy on Digital Health 2020–2025, which emphasise accessibility, usability, and equity [15].

### 4.6. Strengths, Limitations, and Research Priorities

Strengths of the study include the use of validated HLS19 instruments and the combined assessment of digital and navigational literacy across elective and emergency surgical contexts. However, the single-centre, cross-sectional design limits generalisability. Sample sizes within specific subgroups were modest, reducing statistical power, and missing data—especially in the NAV scale—may reflect item difficulty and introduce bias. The absence of contextual variables such as socioeconomic status or prior digital experience may also have limited explanatory depth.

Future research should prioritise multicentre and longitudinal studies, incorporate mixed-methods approaches to explore patient experiences, evaluate literacy-sensitive interventions (including caregiver-mediated support), and develop user-centred digital tools designed to enhance both digital and navigational competencies.

## 5. Conclusions

This study provides updated and methodologically robust evidence on digital and navigational health literacy among surgical inpatients. While descriptive analyses suggested age-related differences—echoing prior studies indicating lower digital engagement among older adults—age was not an independent predictor of either literacy domain after adjusting for sociodemographic and clinical variables. Instead, digital literacy was independently associated only with employment status, with self-employed patients showing significantly lower digital health literacy.

By contrast, navigational literacy was strongly shaped by contextual factors, with patients undergoing emergency procedures presenting substantially lower navigational literacy even after full adjustment. These findings refine earlier assumptions that traditionally vulnerable groups, particularly older adults, would present the most significant limitations; instead, they highlight that vulnerabilities during surgical transitions are more closely linked to the acute, high-pressure nature of emergency care than to age alone.

As digital health systems evolve toward digital-first models, it is essential that care integrates literacy-sensitive communication strategies, simplified and multimodal formats, and maintains parallel non-digital pathways—particularly for patients undergoing emergency surgery or those at risk of digital exclusion. Health professionals, especially nurses, play a key mediating role and should adopt tailored educational strategies that strengthen patient understanding, self-management, and safe transitions from hospital to home.

Despite its contributions, the study’s single-centre design and limited sample size—particularly within specific subgroups—restrict the generalisability of the findings. Consequently, the observed patterns should be interpreted as preliminary rather than definitive. Future research should prioritise multicentre and longitudinal designs, investigate the mechanisms linking surgical context to literacy-related barriers, and develop targeted interventions to support patients facing digital and navigational challenges. Ensuring that health literacy is systematically embedded into digital transformation policies, in alignment with WHO digital health strategies, will be crucial so that technological innovation reduces—rather than widens—existing inequalities and promotes equitable, person-centred care.

## Figures and Tables

**Figure 1 healthcare-13-03227-f001:**
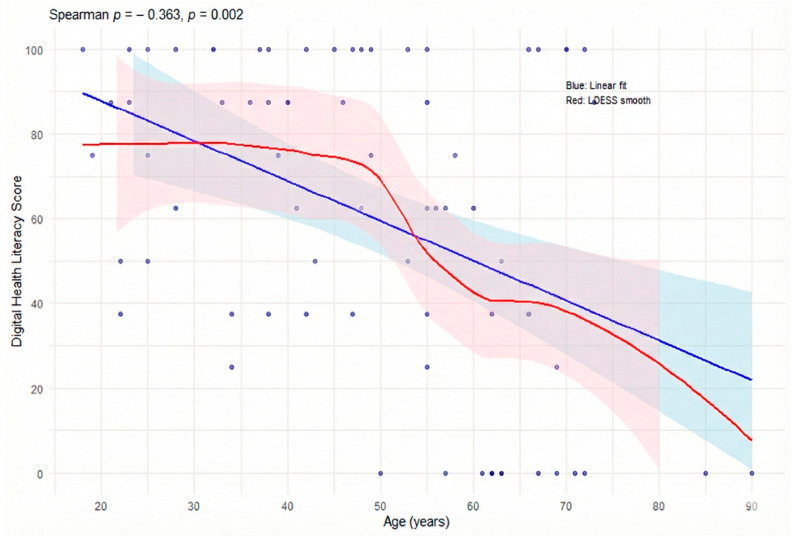
Correlation Between Age and Digital Health Literacy.

**Figure 2 healthcare-13-03227-f002:**
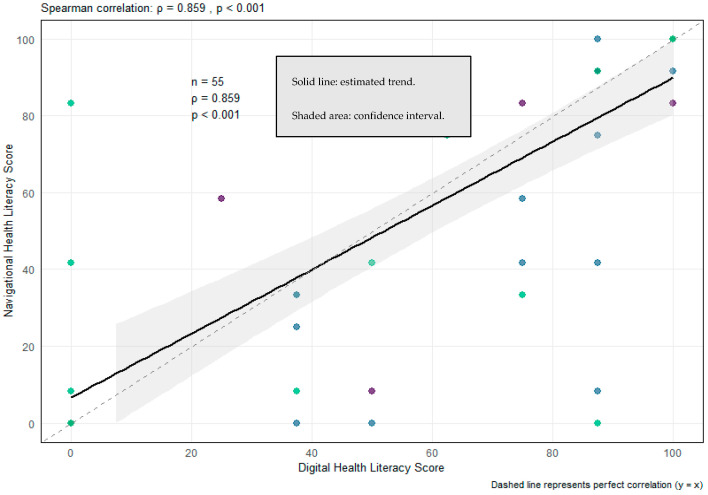
Relationship Between Digital and Navigational Health Literacy.

**Table 1 healthcare-13-03227-t001:** Sample Characteristics (*n* = 94). Continuous variable reported as median (IQR: P25–P75); categorical variables as *n* (%).

Characteristic	*n* (%) or Mean ± SD
Sociodemographic	
Age, years	52.5 (38–63)
Age groups	
- 18–30 years	14 (14.9)
- 31–45 years	22 (23.4)
- 46–50 years	9 (9.6)
- 51–65 years	27 (28.7)
- 65+ years	22 (23.4)
Female sex	37 (39.4)
Male sex	57 (60.6)
Education level	
- None/Primary (1st to 4th grade)	37 (39.4)
- Basic education (5th to 9th grade)	34 (36.2)
- Secondary (10th–12th grade)/Higher	20 (21.3)
Employment status	
- Employed	57 (60.6)
- Self-employed	14 (14.9)
- Unemployed	4 (4.3)
- Retired	14 (14.9)
- Student	3 (3.2)
- Other	2 (2.1)
Surgery context	
- Elective	71 (75.5)
- Emergency	23 (24.5)
Healthcare utilization	
- Rarely	61 (64.9)
- Monthly	30 (31.9)
Has digital support	
-Yes	86 (91.5)
-NO	8 (8.5)

**Table 2 healthcare-13-03227-t002:** **Digital health literacy (HLS19-DIGI) by participant characteristics (median, IQR)**. Values are median (P25–P75). *p*-values from Mann–Whitney U (2 groups) or Kruskal–Wallis (>2 groups).

Predictor	Group	N	Median (IQR)	*p*-Value
Age group	18–30	12	81.2 (59.4–100.0)	0.038
	31–45	18	87.5 (53.1–100.0)	
	46–50	8	81.2 (56.2–100.0)	
	51–65	19	50.0 (0.0–62.5)	
	>65	14	31.2 (31.2–93.8)	
Sex	Female	27	62.5 (31.2–93.8)	0.531
	Male	44	68.8 (37.5–100.0)	
Education	None/Primary (1st to 4th grade)	30	62.5 (37.5–87.5)	0.188
	Basic (5th to 9th grade)	28	87.5 (50.0–100.0)	
	Secondary (10th–12th grade)/Higher	11	0.0 (0.0–100.0)	
Employment	Employed	42	75.0 (50.0–100.0)	0.095
	Self-employed	12	37.5 (0.0–78.1)	
	Unemployed	4	62.5 (46.9–68.8)	
	Retired	10	12.5 (0.0–96.9)	
	Student	2	93.8 (90.6–96.9)	
	Other	1	37.5 (37.5–37.5)	
Surgery context	Elective	53	87.5 (37.5–100.0)	0.097
	Emergency	18	43.8 (37.5–75.0)	
Digital support	Yes	65	62.5 (37.5–100.0)	0.966
	No	6	68.8 (31.2–96.9)	

**Table 3 healthcare-13-03227-t003:** Navigational health literacy (HLS19-NAV) by participant characteristics (median, IQR).

Predictor	Group	N	Median (IQR)	*p*-Value
Age group	18–30	7	91.7 (83.3–100.0)	0.239
	31–45	14	83.3 (45.8–97.9)	
	46–50	8	62.5 (20.8–100.0)	
	51–65	19	41.7 (8.33–83.3)	
	>65	10	91.7 (8.33–100.0)	
Sex	Female	25	58.3 (8.33–100.0)	0.397
	Male	33	83.3 (33.3–100.0)	
Education	None/Primary (1st to 4th grade)	23	75.0 (33.3–87.5)	0.582
	Basic (5th to 9th grade)	24	87.5 (39.6–100.0)	
	Secondary (10th–12th grade)/Higher	9	25.0 (0.0–100.0)	
Employment	Employed	36	83.3 (33.3–100.0)	0.425
	Self-employed	9	41.7 (8.33–41.7)	
	Unemployed	3	83.3 (41.7–87.5)	
	Retired	8	45.8 (100.0–100.0)	
	Student	1	100.0 (0.0)	
	Other	1	8.33 (8.33–8.33)	
Surgery context	Elective	41	83.3 (25.0–100.0)	0.036
	Emergency	17	41.7 (8.33–58.3)	
Digital support	Yes	53	83.3 (25.0–100.0)	0.789
	No	5	58.3 (0.0–100.0)	

**Table 4 healthcare-13-03227-t004:** Multiple linear regression models for digital health literacy (DIGI).

Variable	β (Estimate)	SE	95% CI	*p*-Value
Intercept	110.52	16.41	77.64 to 143.40	<0.001
Age (years)	−0.62	0.38	−1.39 to 0.15	0.111
Female (vs. Male)	2.55	8.90	−15.29 to 20.40	0.776
None/Primary (vs. Basic)	−9.43	10.04	−29.54 to 10.69	0.352
Secondary/Higher (vs. Basic)	18.69	18.07	−17.53 to 54.91	0.306
Self-employed (vs. Employed)	−36.06	13.32	−62.75 to −9.36	0.009
Unemployed (vs. Employed)	−20.68	18.59	−57.94 to 16.58	0.271
Retired (vs. Employed)	−26.92	18.11	−63.22 to 9.38	0.143
Student (vs. Employed)	37.75	32.84	−28.06 to 103.56	0.255
Other (vs. Employed)	−14.22	36.85	−88.07 to 59.63	0.701
Emergency (vs. Elective)	−16.18	10.28	−36.78 to 4.42	0.121
Monthly (vs. Rarely)	−6.63	10.41	−27.49 to 14.23	0.527
Weekly/Daily (vs. Rarely)	3.67	26.14	−48.71 to 56.04	0.889
No (vs. Yes) digital support	−24.88	19.13	−63.21 to 13.46	0.199

Model fit R^2^ = 0.334, adjusted R^2^ = 0.177; F(13,55) = 2.12, *p* = 0.027. Reference categories: sex = Male; education = Basic; employment = Employed; surgery context = Elective; health service use = Rarely; digital support = Yes.

**Table 5 healthcare-13-03227-t005:** Multiple linear regression models for navigation health literacy (NAV).

Variable	β (Estimate)	SE	95% CI	*p*-Value
Intercept	110.47	22.49	65.07 to 155.86	<0.001
Age (years)	−0.58	0.50	−1.59 to 0.43	0.253
Female (vs. Male)	−6.80	10.79	−28.58 to 15.00	0.532
Primary/None (vs. Basic)	−3.74	12.39	−28.75 to 21.26	0.764
Secondary/Higher (vs. Basic)	13.49	22.09	−31.09 to 58.08	0.545
Self-employed (vs. Employed)	−26.27	16.60	−59.76 to 7.22	0.121
Unemployed (vs. Employed)	−21.14	24.20	−69.96 to 27.69	0.387
Retired (vs. Employed)	−26.22	23.33	−73.30 to 20.85	0.267
Student (vs. Employed)	57.69	46.63	−36.42 to 151.80	0.223
Other (vs. Employed)	−35.14	42.14	−120.17 to 49.90	0.409
Emergency (vs. Elective)	−31.07	12.60	−56.49 to −5.66	0.018
Monthly (vs. Rarely)	6.85	14.13	−21.66 to 35.37	0.630
Weekly/Daily (vs. Rarely)	−33.97	25.47	−85.37 to 17.44	0.190
No (vs. Yes) digital support	−22.90	21.45	−66.19 to 20.40	0.292

Model fit R^2^ = 0.312, adjusted R^2^ = 0.099; F(13,42) = 1.46, *p* = 0.172. Reference categories: sex = Male; education = Basic; employment = Employed; surgery context = Elective; health service use = Rarely; digital support = Yes.

## Data Availability

The data supporting the results presented in this study are available from the corresponding author upon request.

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
