# Peer review of "Digital and Navigational Health Literacy in Surgical Patients: Vulnerabilities in the Transition to Post-Discharge Care"

_healthcare, 2025, doi:10.3390/healthcare13243227_

Round 1

Reviewer 1 Report

Comments and Suggestions for Authors

The study highlights significant and timely issue with objective to assess digital and navigational health literacy among surgical patients. The study used validated instruments with appropriate statistical analyses. The findings are well presented and highlight key vulnerable groups, particularly older adults and emergency surgery patients. Nevertheless, the study’s small sample size and very limited representation of older adults reduce the strength and generalizability of the conclusions. Some important characteristics of participant such as income and prior digital exposure are missing, limiting contextual understanding. Additionally, the interpretation of digital illiteracy and the scoring framework require clearer explanation. Overall, the study is relevant and methodologically sound. Please find below some key points to address:

  1. Data collection dates are given as Aug 2025, which shows if the whole study afterward concluded in hurry.
  2. Small sample, limits generalizability, reduces statistical power, and weakens conclusions; especially subgroup analyses (e.g., patients ≥75 years were only n = 5).
  3. No justification for selecting this specific hospital, nor clarity on sampling technique. Please provide justification why convenience sampling was used.
  4. What is the difference between primary education and basic education mentioned under the variable Educational level?
  5. Important variables (socioeconomic status, digital access, prior technology use) are not presented but are directly relevant to digital literacy
  6. Statement “all ≥75 years were digitally illiterate (0%)” seems absolute and may be misleading unless clearly discussed with reasons.
  7. Associations cannot establish causality; this should be mentioned in the limitation.
  8. The conclusion implies generalizability to all surgical patients, but the small, single-center sample limits that.
  9. It seems that findings are not completely aligned with the objectives. Please present findings according to the objectives.
  10. It is not clear why some of the results are presented in analysis section of methodology.

Author Response

  1. The data collection dates are reported as August 2025, which may suggest that the entire study was completed hastily thereafter.
    We would like to clarify that data collection occurred continuously between March and August 2025. Data analysis was conducted later, between September and October 2025. Thus, the study was not carried out hastily.
    The time frame was planned in advance and was not related to any subsequent time pressure. We included this clarification in the “Procedures” section to enhance methodological transparency.

  2. Small samples limit generalizability, reduce statistical power, and weaken conclusions, especially in subgroup analyses (e.g., patients aged ≥75 years represented only n = 5).
    We fully agree. The limitation related to the sample size—particularly for the ≥75 age group—was explicitly added to the Limitations section, highlighting that the small number reduces statistical power, increases imprecision of estimates, and prevents robust generalizations. We emphasised that these results should be interpreted with caution.

  3. There is no justification for selecting this specific hospital, nor clarity regarding the sampling technique. Please justify the use of convenience sampling.
    We revised the “Procedures” section to clarify that:
    the hospital was selected based on operational feasibility, prior ethical approval, and access to the surgical population;
    convenience sampling was used due to logistical constraints of the surgical service and the exploratory nature of the study;
    we acknowledge this approach as a limitation and have added this explanation to the Limitations section.

  4. What is the difference between primary education and basic education as mentioned in the education variable?
    We clarified in the manuscript that:
    Primary education refers to the 1st cycle (1st–4th grade);
    Basic education refers to the 2nd and 3rd cycles (5th–9th grade).
    This explanation has been included in the sociodemographic variables section.

  5. Important variables (socioeconomic status, digital access, prior technology use) are not presented, yet they are directly relevant to digital literacy.
    We appreciate this comment and acknowledge their theoretical relevance to digital literacy. We included in the Limitations section that the absence of these variables limits a more comprehensive understanding of the determinants of digital literacy and should be addressed in future research.

  6. The statement that “all individuals aged 75 and older were digitally illiterate (0%)” seems absolute and may be misleading unless clearly discussed and justified.
    We accept this criticism. We revised the text to emphasize that:
    this conclusion applies only to our very small subgroup (n = 5);
    the lack of variability is due to the extremely small sample and does not allow population-level inferences;
    this observation is presented as a descriptive, non-generalizable finding.
    This clarification was added to both the Results and Discussion sections.

  7. Associations do not establish causality; this should be mentioned in the limitations.
    We explicitly added to the Limitations section that, as a cross-sectional observational study, causal relationships cannot be established between variables. All interpretations are strictly correlational.

  8. The conclusion implies generalization to all surgical patients, but the small sample and single-center design limit such generalization.
    We revised the conclusion to:
    • emphasize that the results apply to the studied sample;
    • acknowledge that the single-center design and limited sample size restrict generalizability;
    • highlight that the findings suggest trends that should be examined in multicenter studies with larger samples.

  9. It is unclear why some results are not fully aligned with the study objectives. Please present results according to the objectives.
    We revised the Results section to align it clearly with the four predefined study objectives:
    3.2 Health Literacy Levels addresses Objective 1
    3.3 Age-Related Disparities, 3.4 Clinical Predictors, and 3.6 Multivariable Analysis address Objective 2
    3.5 Relationship Between Digital and Navigational Literacy addresses Objective 3
    3.7 Vulnerable Subgroup Analysis addresses Objective 4

  10. It is not clear why some results appear in the methods analysis section.
    We agree with this observation. We removed the results that had been inadvertently included in the Methods section and relocated them to the appropriate Results section, ensuring structural coherence in accordance with STROBE guidelines.

Reviewer 2 Report

Comments and Suggestions for Authors

Comments: "Digital and Navigational Health Literacy in Surgical Patients"

Thank you to the authors for their innovative and well-constructed study examining digital and navigational health literacy in surgical patients. The investigation of vulnerabilities during the transition to post-discharge care represents important work in the field. While the study shows significant promise, I recommend major revisions to the methodology section before publication, as outlined below.

Major Methodological Revisions Required

1. Measurement Instruments

Please provide more detailed information about the Digital Health Literacy (HLS19-DIGI) and Navigational Health Literacy (HLS19-NAV) instruments. I recommend adding supplementary questions under each instrument as an appendix to enhance reproducibility and clarity.

2. Statistical Analysis

Given that both primary outcomes are continuous variables with non-normal distributions (expected with smaller sample sizes), the following analytical approach is mandatory:

a. Outcome Reporting

  • Express outcomes as medians with interquartile ranges (IQR) rather than means with standard deviations.

b. Bivariate Analysis

  • Implement appropriate non-parametric tests:
    • Mann-Whitney U tests for binary categorical variables
    • Kruskal-Wallis tests for categorical variables with more than two levels

c. Required Tables

  • Table 2: Digital Health Literacy Scores by Participant Characteristics

    • Display median scores with IQR for each demographic group
    • Include appropriate statistical test results (p-values)
    • Cover all predictors from Table 1
  • Table 3: Navigational Health Literacy Scores by Participant Characteristics

    • Format similarly to Table 2, showing median scores with IQR
    • Include p-values from non-parametric tests

3. Demographic Variable Categorization

  • Age: Recategorize as 18-30, 31-45, 46-50, 51-65, and >65 years to address small sample sizes in certain categories
  • Gender: Ensure clear male/female categories are reported
  • Employment: Revise "other" category; use specific categories such as "student," "not working," "working"
  • Digital support: Reformat as a binary yes/no variable

4. Regression Analysis and Model Specification ( Regression Analysis)

For multiple regression analysis, evaluate whether log transformation or other transformations improve the distribution of the outcome variables. If transformation does not significantly improve model fit or assumption compliance, proceed with non-transformed data and clearly report this decision along with diagnostic test results. Ensure all regression assumptions are tested and reported regardless of transformation decision.

  • Variable Transformation and Model Selection:

    • Evaluate whether log transformation or other transformations improve the distribution of the outcome variables
    • If transformation does not significantly improve model fit or assumption compliance, proceed with non-transformed data

  • Categorical Variable Coding:

    • Include all categorical variables as dummy variables in the regression model
    • For Education variable: Secondary education (largest group) = 0 (reference), Primary education = 1, University education = 2
    • For Age categories: 18-30 = 0 (reference, if largest group), 31-45 = 1, 46-50 = 2, 51-65 = 3, >65 = 4
    • For Gender: Male = 0 (reference), Female = 1
    • For Employment status: Working = 0 (reference, if largest group), Student = 1, Not working = 2
    • For Digital support: No = 0 (reference), Yes = 1
  • Model Building:

    • Include all predictors from Table 1 in the regression models
    • Test for multicollinearity using variance inflation factors (VIF)
    • If multicollinearity is present (VIF > 5), clearly document which variables were removed and provide statistical justification
    • Present a complete regression table with all included variables, coefficients, standard errors, p-values, and 95% confidence intervals

Discussion Section Improvements

Discussion requires enhancement in three key areas:

  1. Expand result interpretation with relevant references to contextualize findings.

  2. Add future research section addressing sample size limitations, suggesting longitudinal designs, mixed-methods approaches, and validation studies in diverse surgical populations.

  3. Include policy implications section on addressing digital literacy gaps, interventions for vulnerable patients, and healthcare system recommendations.

Conclusion

The authors have conducted a valuable study addressing an important topic in surgical patient care. However, the methodological improvements outlined above are mandatory to ensure scientific rigor and enhance the validity of findings. I look forward to reviewing the revised manuscript with these methodological and discussion enhancements.

Author Response

1. Measurement Instruments

Please provide more detailed information about the Digital Health Literacy (HLS19-DIGI) and Navigational Health Literacy (HLS19-NAV) instruments. I recommend adding supplementary questions under each instrument as an appendix to enhance reproducibility and clarity.

Authors’ Response:
We thank the reviewer for the comment. The manuscript has been updated to include a more detailed description of the structure, domains, validation processes, and psychometric properties of the HLS19-DIGI and HLS19-NAV instruments. Additionally, we have included in Appendix A the full list of items used in the HLS19-DIGI and in Appendix B the complete set of items used in the HLS19-NAV.

2. Statistical Analysis

Given that both primary outcomes are continuous variables with non-normal distributions (expected with smaller sample sizes), the following analytical approach is mandatory:

a. Outcome Reporting

  • Express outcomes as medians with interquartile ranges (IQR) rather than means with standard deviations

Authors’ Response:
We thank the reviewer for the comment. The revised version of the manuscript now clarifies that, due to non-normal distributions, the results are reported as medians with interquartile ranges (IQR). This is described in Sections 3.1 and 3.2 of the Results.

b. Bivariate Analysis

  • Implement appropriate non-parametric tests:
    • Mann-Whitney U tests for binary categorical variables
    • Kruskal-Wallis tests for categorical variables with more than two levels

Authors’ Response:
We thank the reviewer for the comment. The manuscript specifies the use of Mann–Whitney U tests, Kruskal–Wallis tests, Dunn post-hoc tests with Bonferroni correction, and Spearman correlations (with bootstrap). These procedures are presented in Section 2.5 Statistical Analysis.

c. Required Tables

  • Table 2: Digital Health Literacy Scores by Participant Characteristics

    • Display median scores with IQR for each demographic group
    • Include appropriate statistical test results (p-values)
    • Cover all predictors from Table 1
  • Table 3: Navigational Health Literacy Scores by Participant Characteristics

    • Format similarly to Table 2, showing median scores with IQR
    • Include p-values from non-parametric tests

Authors’ Response:
We thank the reviewer for the comment. Tables 2 and 3 have been adjusted in the final version to meet the requirements, presenting medians, interquartile ranges (IQR), p-values, and all predictors listed in Table 1, as shown in Sections 3.3 and 3.4.

3. Demographic Variable Categorization

  • Age: Recategorize as 18-30, 31-45, 46-50, 51-65, and >65 years to address small sample sizes in certain categories
  • Gender: Ensure clear male/female categories are reported
  • Employment: Revise "other" category; use specific categories such as "student," "not working," "working"
  • Digital support: Reformat as a binary yes/no variable

Authors’ Response:
We thank the reviewer for the comment. Age has been recategorized into groups (Section 2.3.3 and Table 1), gender is now presented clearly as a binary variable, employment categories have been reorganized (employed, retired, self-employed, etc.), and digital support is described as a binary yes/no variable.

4. Regression Analysis and Model Specification ( Regression Analysis)

For multiple regression analysis, evaluate whether log transformation or other transformations improve the distribution of the outcome variables. If transformation does not significantly improve model fit or assumption compliance, proceed with non-transformed data and clearly report this decision along with diagnostic test results. Ensure all regression assumptions are tested and reported regardless of transformation decision.

  • Variable Transformation and Model Selection:

    • Evaluate whether log transformation or other transformations improve the distribution of the outcome variables
    • If transformation does not significantly improve model fit or assumption compliance, proceed with non-transformed data

  • Categorical Variable Coding:

    • Include all categorical variables as dummy variables in the regression model
    • For Education variable: Secondary education (largest group) = 0 (reference), Primary education = 1, University education = 2
    • For Age categories: 18-30 = 0 (reference, if largest group), 31-45 = 1, 46-50 = 2, 51-65 = 3, >65 = 4
    • For Gender: Male = 0 (reference), Female = 1
    • For Employment status: Working = 0 (reference, if largest group), Student = 1, Not working = 2
    • For Digital support: No = 0 (reference), Yes = 1
  • Model Building:

    • Include all predictors from Table 1 in the regression models
    • Test for multicollinearity using variance inflation factors (VIF)
    • If multicollinearity is present (VIF > 5), clearly document which variables were removed and provide statistical justification
    • Present a complete regression table with all included variables, coefficients, standard errors, p-values, and 95% confidence intervals

Resposta dos autores:

Agradecemos o comentário. Foi revista a  secção 2.5 de forma a descrever a normalidade testada, rejeição de transformações por não melhorar o modelo, criação de dummies (por ex. emergência = 1), avaliação de VIF (<2), verificação de resíduos, listwise deletion eModelos completos incluídos. A tabela 2 apresenta modelos completos com β, SE, IC95%, p e estatísticas de ajuste.

Discussion Section Improvements

Discussion requires enhancement in three key areas:

  1. Expand result interpretation with relevant references to contextualize findings.

  2. Add future research section addressing sample size limitations, suggesting longitudinal designs, mixed-methods approaches, and validation studies in diverse surgical populations.

  3. Include policy implications section on addressing digital literacy gaps, interventions for vulnerable patients, and healthcare system recommendations.

Authors’ Response:
We thank the reviewer for the comment. The revised version fully meets these requirements:
– Sections 4.1–4.5 provide contextualisation using international literature (Sørensen, WHO, M-POHL, etc.);
– Section 4.6 includes recommendations for future studies (longitudinal, multicentre, mixed-methods);
– Section 4.5 and the Conclusions address policy implications for the national health system, digital inclusion, and organisational health literacy.

Conclusion

The authors have conducted a valuable study addressing an important topic in surgical patient care. However, the methodological improvements outlined above are mandatory to ensure scientific rigor and enhance the validity of findings. I look forward to reviewing the revised manuscript with these methodological and discussion enhancements.

Authors’ Response:
We thank the reviewer for the positive appraisal of our work and for the constructive methodological recommendations provided. All suggested improvements have been carefully addressed to strengthen the scientific rigor and validity of the study. The revised manuscript now incorporates the required methodological clarifications, expanded statistical reporting, enhanced discussion, and a more detailed examination of limitations and implications. 

Reviewer 3 Report

Comments and Suggestions for Authors

Thanks for the interesting read. The study employs recently developed, validated scales (HLS19-DIGI and HLS19-NAV) to measure digital and navigational health literacy. Also, we beleive theat the research design and analytical methods are generally appropriate for the study aims. The authors clearly state it was a cross-sectional observational study following STROBE guidelines. However, the use of a cross-sectional design and convenience sampling limits the robustness of the findings.

We think that the Authors sould note the lack of stratification and limited generalisability as a methodological limitation, but the impact could be discussed in greater depth.

Regading data issues, the manuscript reveals an important issue with missing data, particularly for the navigational health literacy scale. The Results note a 38.3% non-completion rate for the HLS19-NAV instrument – meaning over a third of participants could not or did not finish answering those items, compared to 24.5% missing on the digital scale. This fact could lead to bias of the results.

Overall, the manuscript is well-organised and clearly written with some minor improvements needed.

Author Response

Authors’ Response:
We thank the reviewer for the insightful and constructive comments. In the revised manuscript, we have strengthened the methodological discussion to explicitly acknowledge the limitations associated with the cross-sectional design and the use of convenience sampling. We now emphasise that the absence of stratification and the non-probabilistic nature of the sample constrain representativeness and limit the generalisability of the findings to broader surgical populations.

We also expanded the discussion on the impact of small subgroup sizes—particularly for participants aged ≥75 years—which reduces the precision of estimates and further restricts the robustness of subgroup comparisons. Additionally, we addressed the important issue of missing data, noting the high non-completion rate for the HLS19-NAV instrument (38.3%) relative to the digital scale (24.5%). As highlighted by the reviewer, this level of missingness may introduce bias and potentially underestimate the extent of navigational health literacy difficulties. These points have been incorporated into the revised limitations section, where we clarify how missing data and sample characteristics may influence the interpretation of results.

Authors’ Response:
We thank the reviewer for the insightful and constructive comments. In the revised manuscript, we have strengthened the methodological discussion to explicitly acknowledge the limitations associated with the cross-sectional design and the use of convenience sampling. We now emphasise that the absence of stratification and the non-probabilistic nature of the sample constrain representativeness and limit the generalisability of the findings to broader surgical populations.

We also expanded the discussion on the impact of small subgroup sizes—particularly for participants aged ≥75 years—which reduces the precision of estimates and further restricts the robustness of subgroup comparisons. Additionally, we addressed the important issue of missing data, noting the high non-completion rate for the HLS19-NAV instrument (38.3%) relative to the digital scale (24.5%). As highlighted by the reviewer, this level of missingness may introduce bias and potentially underestimate the extent of navigational health literacy difficulties. These points have been incorporated into the revised limitations section, where we clarify how missing data and sample characteristics may influence the interpretation of results.

Round 2

Reviewer 2 Report

Comments and Suggestions for Authors

Thank you for the revisions made so far. I appreciate the effort invested in improving the manuscript. However, several important issues must still be addressed before the paper can be accepted. For your tables, regression, and analysis, please ensure full consistency with the methodology and presentation standards used in the referenced paper (Frontiers in Public Health, DOI: 10.3389/fpubh.2023.1265707). This includes the formatting of descriptive statistics, the structure of all tables, and the correct application and reporting of regression analyses.(https://www.frontiersin.org/journals/public-health/articles/10.3389/fpubh.2023.1265707/full)

  1. Table 1 format
    Please change all descriptive statistics in Table 1 to median (IQR: P25–P75).
    In addition, all tables throughout the manuscript should consistently present values in the format: median (IQR: P25–P75).

  2. Multiple linear regression table
    The current regression analysis is incomplete. Not all variables are included, and dummy variables with a clearly defined reference category are not used. This is a critical methodological issue.
    I strongly recommend consulting a biostatistician and redoing the full regression model to ensure:

    • inclusion of all relevant variables,
    • correct use of dummy coding, and
    • clear identification of reference groups.
  3. Discussion section
    Please ensure that the discussion reflects the updated results only. Some parts still cite older results that no longer apply (for example, references to age categories such as “>75 years”). Please revise the discussion carefully so it aligns fully with the revised analyses and tables.

I look forward to receiving the updated version addressing these issues.

Author Response

Dear Reviewer,

We sincerely thank you for your second-round evaluation and for the clarity and precision of your methodological recommendations. We have now implemented all requested revisions, aligning the manuscript fully with the analytical and reporting standards of the article you referenced (Frontiers in Public Health, 2023; 11:1265707).

Below we describe, point by point, how each issue has been addressed in the revised manuscript.

  1. Table 1 — Median (IQR: P25–P75) format

Reviewer comment:
“Please change all descriptive statistics in Table 1 to median (IQR: P25–P75). All tables throughout the manuscript should consistently present values in the format: median (IQR: P25–P75).”

Response:
We have fully revised Table 1. Age is now presented as median (IQR: P25–P75). The table structure was reviewed to ensure complete consistency, using:

n (%) or median (IQR: P25–P75)
as the standard formatting rule.
Tables 2 and 3 have also been harmonised to use “median (P25–P75)” throughout.
The text in Section 3 was updated accordingly to ensure descriptive consistency.

  1. Multiple linear regression — full re-specification, dummy coding, and clearer reporting

Reviewer comment:
“The current regression analysis is incomplete. Not all variables are included, and dummy variables with a clearly defined reference category are not used.”

Response:
We completely rebuilt the regression models following your guidance and in accordance with the referenced Frontiers article. All relevant variables were included:

  • age (continuous)
  • sex (Male/Female)
  • education (Primary/None; Basic; Secondary/Higher)
  • employment (Employed; Self-employed; Unemployed; Retired; Student; Other)
  • surgery context (Elective/Emergency)
  • healthcare utilisation (Rarely; Monthly; Weekly/Daily)
  • digital support (Yes/No)

All categorical predictors were dummy-coded with explicit reference categories, which are now clearly displayed in Table 4 and described in Section 2.5.

The regression models were completely re-estimated.
The updated Table 4 now reports:

  • β coefficients
  • standard errors
  • 95% confidence intervals
  • p-values
  • model fit indices (R², adjusted R², F and p-values)
  • explicit reference category notes

Updated findings:

  • Digital literacy (DIGI): The only independent predictor was employment status, with self-employed participants scoring significantly lower (β = −36.06, p = 0.009). Age and education were not significant.
  • Navigational literacy (NAV): The only independent predictor was emergency surgery (β = −31.07, p = 0.018), even after adjusting for all covariates.

These results replace all previously reported adjusted associations.
The revised analysis is now presented in Section 3.6 and Table 4.

  1. Discussion — fully aligned with updated findings

Reviewer comment:
“Please ensure that the discussion reflects the updated results only. Some parts still cite older results that no longer apply.”

Response:
We have thoroughly revised the Discussion to align with the new regression findings:

  • Removed references to previous age categories (e.g., >75 years).
  • Replaced all outdated interpretations.
  • Emphasised that age differences observed in descriptive analyses were not independent predictors after multivariable adjustment.
  • Highlighted the two true independent predictors:
    – Self-employed → lower digital literacy (DIGI)
    – Emergency surgery → lower navigational literacy (NAV)
  • Updated all numerical references to match Table 4 exactly.
  • Revised the “Implications”, “Limitations”, and “Future Research” subsections to reflect the new analytic picture, especially the central importance of contextual (emergency) rather than demographic predictors.

The revised Discussion now accurately represents the updated statistical models and matches the structure of the referenced Frontiers article.

Final statement

We appreciate the reviewer’s careful attention to methodological detail and believe the manuscript has been substantially strengthened as a result. The analyses are now fully aligned with the recommended standards, and the narrative has been updated to reflect only the corrected and validated results.

We hope this revised version meets your expectations and are grateful for your continued guidance.

With best regards,
Paulo Nogueira & Patrícia Valentim
(on behalf of all authors)

Reviewer 3 Report

Comments and Suggestions for Authors

Thanks for the corrections performed. All comments were addressed.

Author Response

Thank you very much.